# Measuring Coding Challenge Competence With APPS

**Dan Hendrycks**[*]
UC Berkeley

**Steven Basart**[*]
UChicago

**Saurav Kadavath**
UC Berkeley

**Mantas Mazeika**
UIUC

**Akul Arora**
UC Berkeley

**Ethan Guo**
UC Berkeley

**Collin Burns**
UC Berkeley

**Samir Puranik**
UC Berkeley

**Horace He**
Cornell

**Dawn Song**
UC Berkeley

**Jacob Steinhardt**
UC Berkeley

## Abstract

While programming is one of the most broadly applicable skills in modern society, it is unclear how well state-of-the-art machine learning models can write code. Despite its importance, there has been surprisingly little work on evaluating code generation, and it can be difficult to assess code generation performance in an accurate and rigorous manner. To meet this challenge, we introduce APPS, a benchmark for code generation. Unlike prior work in more restricted settings, our benchmark measures the ability of models to take an arbitrary natural language specification and generate satisfactory Python code. Similar to how companies assess candidate software developers, we evaluate models by checking their generated code on test cases. Our benchmark includes 10,000 problems, which range from having simple one-line solutions to being substantial algorithmic challenges. We fine-tune large language models on both GitHub and our training set, and we find that the prevalence of syntax errors is decreasing exponentially as models improve. Recent models such as GPT-Neo can pass approximately 20% of the test cases of introductory problems, so we find that machine learning models are now beginning to learn how to code. As the social significance of automatic code generation increases over the coming years, our benchmark can provide an objective measure for tracking advancements.

"Everybody should learn to program a computer, because it teaches you how to think." – *Steve Jobs*

## 1 Introduction

Computer programming can be found in nearly all parts of society. Spanning entertainment, healthcare, education, and more, programming is an extraordinarily general tool with applications that are vast in scope. As computers are becoming more ubiquitous in modern life, rising demand for high-quality code draws an ever-greater number of aspiring programmers to the profession. After years of study to become proficient coders, human experts are are able to convert abstract specifications of diverse cognitive tasks into concrete programs.

In the past few years, large-scale language models have shown promise in generalizing to various cognitive tasks, including linguistic inference (Wang et al., 2019a), commonsense reasoning (Zellers et al., 2019; Huang et al., 2019; Bisk et al., 2019), logical deduction (Liu et al., 2020), mathematics (Polu and Sutskever, 2020; Hendrycks et al., 2021c), and general understanding of multiple domains

---

[*]Equal Contribution.

35th Conference on Neural Information Processing Systems (NeurIPS 2021) Track on Datasets and Benchmarks.

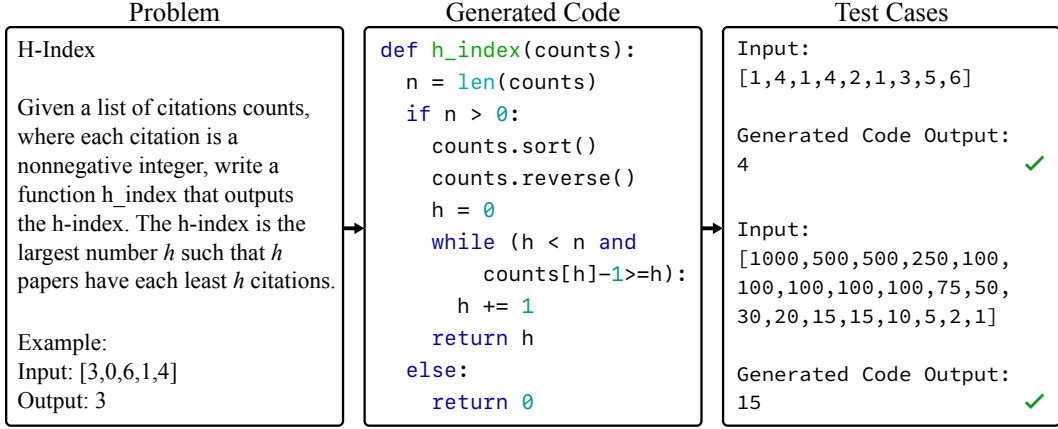

Figure 1: An example problem from APPS (left) along with possible generated code (middle) and two example test cases we use to evaluate the generated code (right). Our evaluation framework has test cases and 10,000 code generation problems of varying difficulty levels.

of human knowledge (Hendrycks et al., 2021b). However, whether large-scale language models can reliably write code remains an open question.

Motivated by the potential of language models and the need for thorough code generation evaluation, we introduce APPS, a benchmark for code generation from natural language specifications. Unlike prior work on code generation with Transformer language models (Vaswani et al., 2017), which mostly focuses on code translation (Lachaux et al., 2020) and pseudocode-to-code (Kulal et al., 2019), we evaluate models on their ability to take specifications given in natural language and write code that meets these specifications. This setting mirrors how human coders are evaluated and is a more realistic and informative setting in which to benchmark models.

APPS provides a precise and comprehensive view of code generation. APPS evaluates models not only on their ability to code syntactically correct programs, but also on their ability to understand task descriptions and devise algorithms to solve these tasks. It contains 10,000 programming problems at various levels of difficulty, covering simple introductory problems, interview-level problems, and coding competition challenges. If a model were to perform well on APPS, this would indicate an ability to flexibly use data structures and programming techniques, as well as an ability to correctly interpret diverse task specifications, follow instructions, and understand human intent (Hendrycks et al., 2021a).

For most text generation tasks, high-quality evaluation requires human feedback, which can be time-consuming or carry pecuniary costs. As a result, automatic metrics such as BLEU (Papineni et al., 2002) are often used to compare methods, but these metrics do not necessarily track program correctness. Since the objective for code generation is to produce correct programs, we assess programs not with BLEU but with test cases and error catching. Evaluating code generation on APPS is facilitated by a large bank of over 130,000 test cases. The test cases are specifically chosen to probe correct functionality across the input space. By using test cases, we provide a gold-standard metric for code generation quality.

In our experiments, we find that models are now starting to exhibit nonzero accuracy and solve some coding problems. Additionally, as models improve, we observe that syntax errors are exponentially decreasing. We also find further evidence that BLEU is a problematic metric for code generation, sometimes being anticorrelated with gold-standard accuracy. We find that accuracy decreases with difficulty level and improves through fine-tuning and model size increases. The strongest model that we evaluate on introductory problems passes almost 20% of test cases given five attempts. These results position code generation as a challenging but now tractable testbed for large-scale language models.

Writing code to meet specifications in natural language is an economically valuable task with widespread social implications should it be solved, as it could eventually facilitate malicious code generation and one day result in job automation. As large-scale language models have the potential

|                          | PY150  | CONCODE    | SPoC       | APPS                 |
|--------------------------|--------|------------|------------|----------------------|
| Programming Language     | Python | Java       | C++        | Python               |
| Test Cases               | ✗      | ✗          | ✓          | ✓                    |
| Number of Programs       | N/A    | 104,000    | 18,356     | 232,421              |
| Lines per Program (Avg.) | 1      | 26.3       | 14.7       | 18.0                 |
| Number of Exercises      | 3,000  | 104,000    | 677        | 10,000               |
| Text Input               | Python | Docstrings | Pseudocode | Problem Descriptions |

Table 1: A comparison of the APPS dataset to existing datasets for converting between text and code. APPS has over an order of magnitude more ground-truth solutions than these datasets, test cases, and natural language problem descriptions.

to make significant progress on code generation, it is essential that we begin to track advancements on this task. Our new benchmark facilitates measuring performance in an accurate and rigorous manner. Using APPS, we find that programming is very difficult for modern language models, though performance is improving. Thus, the APPS benchmark can provide foresight about the performance of future large-scale language models at the critical task of program synthesis from natural language. The dataset is available at https://github.com/hendrycks/apps.

## 2   Related Work

**Program Synthesis.**   Program synthesis is the task of generating a computer program that satisfies given specifications. Deductive program synthesis uses formal logic specifications to define a search problem. Complex optimization techniques are used to generate programs satisfying these specifications (Alur et al., 2018). Because specifications must be converted into a formal language, these approaches can be rigid. Inductive synthesis from example input-output behavior can provide an alternative to formal specification (Cai et al., 2017; Gulwani et al., 2017), but it is often hard to full specify behavior with examples, as any machine learning practitioner is well-aware.

An alternative to formal or inductive specification is to specify program behavior in natural language, which prior work has considered in constrained settings. Raza et al. (2015) and Desai et al. (2016) generate short programs using ad-hoc programming languages to solve specifications such as "Any 2 letters followed by any combination of 6 whole numbers." Yu et al. (2018) introduce the Spider dataset for converting natural language queries into short SQL database commands. In contrast, we consider long natural language specifications and general-purpose programming languages.

**Code Understanding Datasets.**   Language modeling is a compelling tool for code generation, and several works have achieved success generating code with language models in limited settings. Lachaux et al. (2020) use unsupervised machine translation techniques to translate functions across programming languages, attaining identical behavior after translation in many cases. Kulal et al. (2019) introduce SPoC, a method for converting pseudocode to code utilizing seq2seq machine translation with an additional search step. To train SPoC, they collect line-by-line descriptions of C++ programs using Amazon Mechanical Turk. Recently, Lu et al. (2021) introduce the CodeXGLUE benchmark which aggregates various previous benchmarks and use CodeBLEU (Ren et al., 2020) and CONCODE. Iyer et al. (2018) investigate generating Java code from docstrings and evaluate performance with BLEU. The docstrings are often incomplete specifications of what should be coded and only 14.7 words long on average, e.g. "Convert mixed case to underscores." By comparison, problem specifications in our new APPS benchmark are self-contained and have a much larger average length of 293.2 words. Unlike Iyer et al. (2018), APPS contains test cases for every exercise, enabling a high-quality evaluation of code correctness. Further comparisons are in the Appendix.

**Evaluating Large-Scale Language Models.**   Modern large-scale language models have demonstrated impressive capabilities across a variety of text-based tasks. On the SuperGLUE benchmark (Wang et al., 2019b), some models now exceed human performance. On many commonsense reasoning benchmarks, performance is rising quickly (Zellers et al., 2019; Huang et al., 2019; Bisk et al., 2019). Even when language models are evaluated across diverse technical areas such as law and medicine, performance is surprisingly high and poised to improve as models are scaled up further (Hendrycks et al., 2021b). With rapid improvements across numerous datasets, finding resilient

benchmarks on which models significantly underperform humans is challenging. APPS represents an attempt to fill this gap and cleanly separate model performance from that of expert humans.

## 3 The APPS Dataset

The APPS dataset consists of problems collected from different open-access coding websites such as Codeforces, Kattis, and more. The APPS benchmark attempts to mirror how humans programmers are evaluated by posing coding problems in unrestricted natural language and using test cases to evaluate solution correctness. The problems range in difficulty from introductory to collegiate competition level and measure coding and problem-solving ability.

The Automated Programming Progress Standard, abbreviated APPS, consists of 10,000 coding problems in total, with 131,777 test cases for checking solutions and 232,421 ground-truth solutions written by humans. Problems can be complicated, as the average length of a problem is 293.2 words. The data are split evenly into training and test sets, with 5,000 problems each. In the test set, every problem has multiple test cases, and the average number of test cases is 21.2. Each test case is specifically designed for the corresponding problem, enabling us to rigorously evaluate program functionality.

**Dataset Construction.** To create the APPS dataset, we manually curate problems from open-access sites where programmers share problems with each other, including Codewars, AtCoder, Kattis, and Codeforces. Problems are posed as natural language specifications of what should be coded, and they come in various formats. To improve quality and consistency, we wrote custom HTML parsers for each source of problems, which allows us to properly format LaTeX expressions, lists, and sections in the question text. Where necessary, we convert equation images to LaTeX using the MathPix API, and we remove problems that rely on image figures. We also perform deduplication using tf-idf features with SVD dimensionality reduction and cosine similarity. Several graduate and undergraduate student authors polished and refined this dataset over the course of six months, ensuring a high-quality set of problems.

Executing and evaluating arbitrary Python code is challenging. On the websites we source data from, human solutions are allowed to run arbitrary code, including import statements for common modules and libraries. To handle this, each website implements a custom judging system for solutions. We design a testing framework with this in mind, which merges the judging functionality of several websites. We also standardize the format of test cases. The end result is that solutions are allowed to execute arbitrary Python code, and the results are compared against test cases for a given problem.

**Dataset Difficulty.** Each of our problem sources uses a separate scale for measuring difficulty. We place problems from these different sources into three categories. For example, problems from Kattis with difficulty less than 3 are categorized as "introductory," problems with difficulty between 3 and 5 as "interview," and problems with difficulty greater than 5 as "competition."

1. **Introductory Level**. These are problems that most programmers with 1-2 years of experience can answer without requiring complicated algorithms. Examples of such problems include counting the number of vowels in a string, or returning the running sum of a list of integers. There are 3,639 problems classified as introductory level and 1,000 in the test set.

2. **Interview Level**. These are problems that are more algorithmic and difficult in nature and would be at the level of questions asked in programming technical interviews. Examples of such problems might include those involving data structures such as trees or graphs, or problems that requiring nontrivial algorithms. There are 5,000 problems classified as interview level and 3,000 in the test set.

3. **Competition Level**. These are problems are the most challenging and are at the level of the most advanced high school and collegiate programming competitions, including USACO, IOI, and ACM. There are 1,361 competition level problems and 1,000 in the test set.

**Problem Formats.** To accommodate a broad range of problem sources, problems in APPS come in two formats.

- **Call-Based Format** problems generally provide initial starter code, usually in the form of a function header, and ask for the solution to be provided as the function's return value.

**Problem**

You are given a string $s = s_1 s_2 \ldots s_n$ of length $n$, which only contains digits 1, 2,..., 9. A substring $s[l \ldots r]$ of $s$ is a string $s_l s_{l+1} s_{l+2} \ldots s_r$. A substring $s[l \ldots r]$ of $s$ is called even if the number represented by it is even. Find the number of even substrings of $s$. Note, that even if some substrings are equal as strings, but have different $l$ and $r$, they are counted as different substrings. The first line contains an integer $n$ ($1 \leq n \leq 65000$) — the length of the string $s$. The second line contains a string $s$ of length $n$. The string $s$ consists only of digits 1, 2,..., 9. Print the number of even substrings of $s$.

**Model Output**

```
n = int(input())
s = input()
ans = 0
for i in range(n):
    for j in range(i, n):
        if int(s[i:j+1]) % 2 == 0:
            ans += 1
print(ans)
```

Figure 2: Model-generated code from GPT-2 1.5B. 18/18 test cases pass for the problem. Note that models were given in problems exactly as they they are formatted here, which even meant parsing LaTeX. For brevity, we exclude formatting instructions from this depicted problem.

- **Standard Input Format** problems generally lack starter code. Instead, the model is only provided with the problem and must output its answers to the STDOUT stream, such as by using print statements.

For the call-based format problems, we prompt models using the following inputs:

```
"\nQUESTION:\n" + q_str + "\n" + starter_code_str + "\n" + "\nUse
Call-Based Format\n\nANSWER:\n"
```

For the above prompt, the variable `q_str` represents the raw text of the problem statement. The variable `starter_code_str` represents the starter code given in the problem definition, or the empty string if no starter code was provided. For the standard input format problems, we prompt the model with the input string as before, but we replace "Call-Based Format" with "Standard Input Format." Note that if starter code is given, it is only part of the input. This means that to use the starter code, a model must learn to copy the starter code at the beginning of its outputted answer in order to get the question correct. We find that fine-tuned models are able to do this without difficulty.

**Test Case Quality.** In the APPS test split, the average number of test cases is 21.2, but some problems only have two test cases. These problems mainly come from Kattis and were chosen for the test split due to limited numbers of competition problems. A potential concern is that these problems could result in false positives if models happen to guess both test cases correctly. This is very unlikely in problems with large output spaces, but some problems have small output spaces, such as {"YES", "NO"}. Although the model must narrow down these two options from the space of all possible strings, we find that this is not difficult for current models to accomplish, and this may cause false positives in our evaluation.

To quantify the impact of these problems, we took all 2- and 3-test-case problems that GPT-Neo 2.7B passed and manually checked for false positives. Out of 12 problems that the model got correct with 2 test cases, 8 were false positives. Out of 6 problems with 3 test cases, only 1 was a false positive. Since the test cases are from coding challenge websites with many users, we can assume that false negatives are absent. Thus, the false positive rate with 2 test cases is $8/(890 - 4) = 0.009 < 1\%$ and $1/(765 - 5) = 0.001 < 1\%$ with 3 test cases. The upshot is that the potential for noise is significantly less than that of most naturally annotated datasets.

**Problem**

You are given two integers $n$ and $m$. Calculate the number of pairs of arrays $(a, b)$ such that: the length of both arrays is equal to $m$; each element of each array is an integer between 1 and $n$ (inclusive); $a_i \leq b_i$ for any index $i$ from 1 to $m$; array $a$ is sorted in non-descending order; array $b$ is sorted in non-ascending order. As the result can be very large, you should print it modulo $10^9 + 7$. Input: The only line contains two integers $n$ and $m$ ($1 \leq n \leq 1000$, $1 \leq m \leq 10$). Output: Print one integer – the number of arrays $a$ and $b$ satisfying the conditions described above modulo $10^9 + 7$.

**Model Output**

```
n, m = map(int, input().split())
mod = 10**9+7

fact = [1, 1]
factinv = [1, 1]
inv = [0, 1]

for i in range(2, n+1):
    fact.append((fact[-1] * i) % mod)
    inv.append((-inv[mod % i] * (mod//i)) % mod)
    factinv.append((factinv[-1] * inv[-1]) % mod)

def cmb(n, r, mod):
    if (r < 0) or (n < r):
        return 0
    r = min(r, n - r)
    return fact[n] * factinv[r] * factinv[n-r] % mod

print((cmb(n, m, mod) * cmb(m-1, n-1, mod)) % mod)
```

Figure 3: An example from GPT-2 1.5B. Although the code generated passes 0 test cases, it looks plausible at first glance.

# 4 Experiments

## 4.1 Experimental Setup

**Models.** We use GPT-2 (Radford et al., 2019), GPT-3 (Brown et al., 2020), and GPT-Neo (Black et al., 2021) models. The GPT architecture is especially appropriate for text generation because it is autoregressive. However, GPT-2 was not pretrained on code, so we pretrain it on GitHub as described in the next paragraph. Anecdotal evidence indicates that GPT-3 can generate code. To determine the extent of its code generation ability, we use the 'davinci' (Instruct series) model, the largest publicly available model speculated to have 175 billion parameters. Finally, GPT-Neo has an architecture similar to GPT-3, and it was pretrained on the Pile (Gao et al., 2020) which includes GitHub. Unlike GPT-3, GPT-Neo's weights are publicly available, hence we are able to fine-tune it with APPS.

**GPT-2 Pretraining.** Since GPT-2 was trained on natural language and not code, we collected GitHub code to further pretrain GPT-2. GitHub repositories with fewer than one star were filtered out. While Neo's GitHub pretraining data did *not* undergo an APPS data decontamination process, our GPT-2 models are trained on decontaminated data. Specifically, all repositories matching certain keywords that would suggest overlap with common programming exercises were removed. We provide the list of keywords in the Supplementary Materials. We also discard any GitHub code that contains functions with the same signatures as functions in the starter code in many of our APPS problems. This leaves us with 30 GB of Python code. To improve the efficiency of pretraining, we process all Python code in the pretraining dataset by converting from spaces to tabs, which saves the character conversion when running model tokenizers.

**Fine-tuning.** During fine-tuning with APPS, the objective is to predict the entire code solution, given both the English text problem statement and the problem format (call-based format or standard input format). For problems with starter code, we exclude the starter code from the training loss.

| Model | Test Case Average | | | | Strict Accuracy | | | |
|---|---|---|---|---|---|---|---|---|
| | Introductory | Interview | Competitive | Average | Introductory | Interview | Competition | Average |
| GPT-2 0.1B | 5.64 | 6.93 | 4.37 | 6.16 | 1.00 | 0.33 | 0.00 | 0.40 |
| GPT-2 1.5B | 7.40 | 9.11 | 5.05 | 7.96 | 1.30 | 0.70 | 0.00 | 0.68 |
| GPT-Neo 2.7B | 14.68 | 9.85 | 6.54 | 10.15 | 3.90 | 0.57 | 0.00 | 1.12 |
| GPT-3 175B | 0.57 | 0.65 | 0.21 | 0.55 | 0.20 | 0.03 | 0.00 | 0.06 |

Table 2: Average percentage of test cases passed and strict accuracy for each model and difficulty level. All values are percentages. Note '0.1B' indicates the number of model parameters in billions. GPT-3 is a *few-shot* model and not fine-tuned, unlike the other models. GPT-Neo does best and attains approximately 4% strict accuracy on Introductory problems, and for these problems it passes approximately 15% of the test cases.

Across pretraining and fine-tuning, we use the AdamW optimizer (Loshchilov and Hutter, 2019), a batch size of 256, and a weight decay of 0.05. We fine-tune for 10 epochs. We use DeepSpeed and its implementation of the ZeRO optimizer to reduce memory consumption while training large models (Rasley et al., 2020; Rajbhandari et al., 2020). Unless otherwise specified, we use the default HuggingFace generation parameters, except that we use beam search with a beam size of 5. Models are fine-tuned on 8 A100 GPUs.

## 4.2   Metrics

To obtain a comprehensive evaluation of code generation ability, we use the large bank of test cases and ground-truth solutions provided with APPS. Test cases allow for *automatic* evaluation, even though the the space of possible programs can be combinatorially large. Therefore, unlike many other text generation tasks, manual analysis is not necessary. We aggregate the generated code's performance on test cases with two metrics, "test case average" and "strict accuracy."

**Test Case Average.**   We compute the average fraction of test cases passed. Concretely, let the number of problems in the test set be $P$. For a given problem $p$, let the code generated to solve problem $p$ be denoted $\langle \texttt{code}_p \rangle$, and set of test cases for problem $p$ be $\{(x_{p,c}, y_{p,c})\}_{c=1}^{C_p}$. Then the test case average is

$$\frac{1}{P} \sum_{p=1}^{P} \frac{1}{C_p} \sum_{c=1}^{C_p} \mathbb{1}\{\texttt{eval}(\langle \texttt{code}_p \rangle, x_{p,c}) = y_{p,c}\}.$$

Oftentimes, solutions can successfully pass a subset of the test cases but not cover every corner case. This allows for less stringent model evaluation, as strict accuracy may currently obscure model improvements.

**Strict Accuracy.**   Eventually, generated solutions should pass all test cases including corner cases. To compute the strict accuracy which requires programs pass every test case, we run the code generated by the model on every test case of every problem. Strict accuracy is then computed by taking the number of solutions passing every test case divided by the total number of exercises. Using the notation from before, we can write the strict accuracy as $\frac{1}{P} \sum_{p=1}^{P} \prod_{c=1}^{C_p} \mathbb{1}\{\texttt{eval}(\langle \texttt{code}_p \rangle, x_{p,c}) = y_{p,c}\}$. Future research may only use strict accuracy when models become sufficiently capable.

## 4.3   Model Performance Analysis

**Qualitative Output Analysis.**   Models can sometimes generate correct or superficially plausible code. Figure 2 shows code generated by GPT-2 1.5B that passes all test cases. When models do not pass the test cases, sometimes their generated code still appears plausible at first glance. For example, in Figure 3, we see that the 1.5B parameter model generates code that is related to the problem statement and makes a plausible attempt to solve it.

**Test Case Evaluation.**   We show the main results in Table 2. We observe that models are able to generate code that passed some test cases, implying many generated programs are free of syntax errors and can successfully process inputs test cases to produce correct answers. Note that for Introductory questions, GPT-Neo passes approximately 15% of the test cases. We visualize Test Case Average

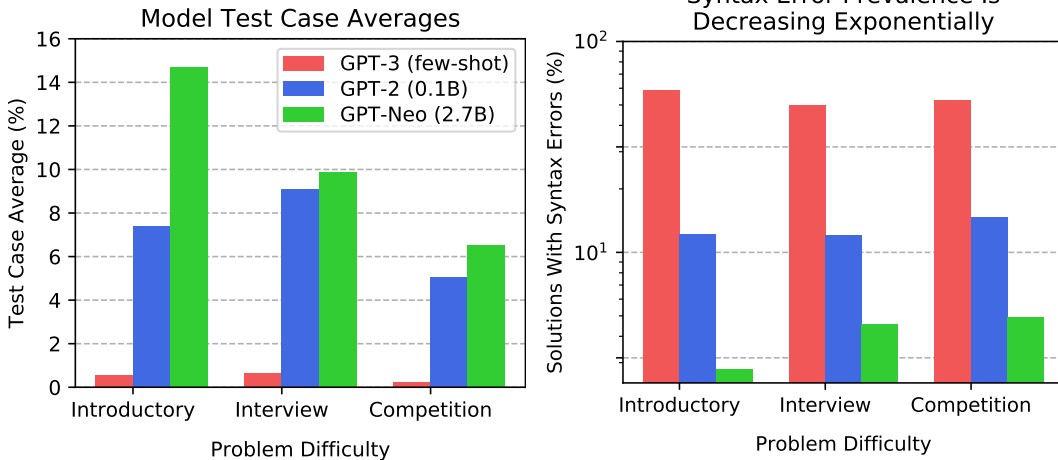

Figure 4: The average percentage of test cases passed increases with larger fine-tuned models.

Figure 5: Syntax errors decrease exponentially with fine-tuning and increased model sizes. GPT-Neo 2.7B has very few syntax errors.

results in Figure 4. This demonstrates models are showing marked improvements on code generation and now starting to have traction on code generation.

Performance can be further improved by sampling multiple solutions and selecting the best. Here, we perform beam search with beam width 5 and evaluate its 5 beams, so that each model has five attempts to get a problem correct rather than one. With this setup, GPT-Neo's strict accuracy on Introductory problem then exceeds 5%, as shown in Table 3. Our results in the Supplementary Materials show that the top-5 test case average GPT-2 0.1B is 10.75 while the top-1 test case average of GPT-2 1.5B is 7.96. This highlights that simply sampling multiple candidate solutions is a powerful way to markedly improve performance.

|  | Top-1 | Top-5 |
|---|---|---|
| Test Case Average | 14.7% | 19.9% |
| Strict Accuracy | 3.9% | 5.5% |

Table 3: GPT-Neo 2.7B performance on introductory problems using one generated program (Top-1) and the best of five generated programs (Top-5). Full results are in the Supplementary Materials.

Our results also provide us with information about the importance of model choice. Evidently existing few-shot GPT-3 models are not necessarily better at code generation than fine-tuned models that are smaller by two orders of magnitude. Additionally, performance improvement from GPT-2 1.5B to GPT-Neo 2.7B is larger than that from GPT-2 0.1B to GPT-2 1.5B. Potential causes of GPT-Neo's better performance are that GPT-Neo is trained on more code from GitHub, it has more parameters, or its architecture hyperparameters were chosen better. Memorization explaining all performance is an implausible explanation as performance tracks problem difficulty; were models just memorizing, we would expect uniform performance across difficulties. Since models still have large room for improvement, solving the APPS benchmark without unreasonable amounts of computational resources may require architectural or algorithmic improvements.

**Syntax Errors.** We now assess the frequency of syntax errors, errors that prevent the program from being interpreted including inconsistent spacing, unbalanced brackets, missing colons, and so on. Syntax errors are identified in our testing framework based on the heuristic of whether pyext is able to load the generated code as a Python module. For our purposes, this almost exclusively occurs for syntax errors. We visualize the prevalence of syntax errors in Figure 5. While approximately 59% of GPT-3's generated solutions for introductory problems have syntax errors, GPT-Neo syntax error frequency is approximately 3%. Note that recent work such as Yasunaga and Liang (2020) create a separate model to repair source code to fix compilation issues, but our results suggest that such efforts may be unnecessary in the future as syntax error frequency is sharply decreasing automatically.

**BLEU.** We find that assessing model performance with BLEU is a poor substitute for evaluating with test cases. To evaluate BLEU, we take the generated solution and compute its BLEU with each human-written solution for a given problem; we then record the highest BLEU score. Observe in Figure 6 that BLEU increases as problem sources become more difficult, even though models actually perform worse on harder problems. Moreover, worse models can have similar or higher BLEU scores. For example, GPT-2 0.1B has 26.8, 29.7, and 30.2 as BLEU scores for introductory, interview, and competition problems, respectively. Meanwhile GPT-Neo 2.7B has 27.1, 29.1, and 29.3 as its BLEU scores, respectively. Hence BLEU wrongly suggests GPT-Neo is a worse model.

**Evaluating GPT-3.** We evaluate GPT-3 175B on APPS in a few-shot setting. A separate prompt is used for standard input and call-based questions, and each prompt includes instruction text along with two example questions and solutions from the corresponding question type. We find that GPT-3 only solves 3 problems out of 5,000: two introductory problems and one interview problem. The two introductory problems are simple interpretation tasks, such as implementing a specified algebraic expression. The interview problem requires higher-level thinking that suggests nontrivial reasoning. However, it is possible that GPT-3 memorized the solution during pretraining, or that it took a lucky guess based on heuristics in the question. One potential factor in GPT-3's poor performance is that it handles syntax poorly. Namely, we observed cases where improper formatting of otherwise functioning code causes a syntax error. For specific examples and more details, see the Supplementary Materials.

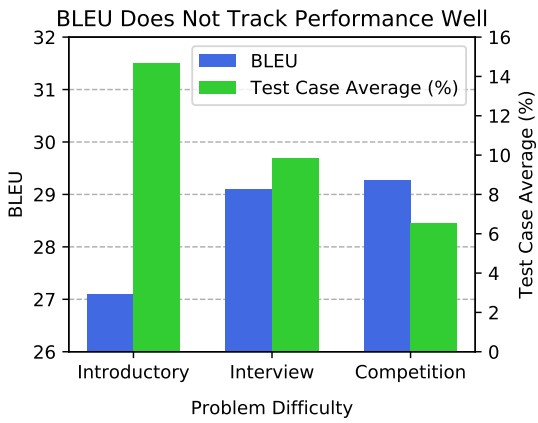

Figure 6: BLEU scores for GPT-Neo 2.7B increase with difficulty level and are anticorrelated with a gold-standard accuracy metric.

**Evaluations on Larger Models.** Since the public release of APPS, several others have trained even larger models on APPS than we evaluate here. OpenAI Codex is a 12B parameter Transformer language model pre-trained on large quantities of public code and comments. Chen et al. (2021) evaluate Codex on APPS under various configurations and achieve top-1 and top-5 accuracy on introductory problems of 4.14% and 9.65% respectively, close to double the top-5 accuracy of GPT-Neo 2.7B. Furthermore, by scaling up to a top-1000 evaluation they obtain 25% accuracy. This demonstrates that larger models trained specifically for code generation can improve APPS performance even further, but are still far from solving the task.

## 5   Conclusion

We introduced APPS, a benchmark of 10,000 Python programming problems. Unlike prior work that focused on pseudocode to code generation or translation between programming languages, our benchmark measures how well language models can generate python code given natural language specifications. By performing extensive quality assurance and including hundreds of thousands of test cases and ground-truth solutions across different difficulty levels, we created a comprehensive and rigorous testbed for evaluating models. We assessed state-of-the-art generative models on our benchmark and found that overall performance was low. However, the prevalence of syntax errors decreased exponentially as models improved, and recent models such as GPT-Neo solved over 5% of our introductory problems. As models become more competent at code generation, it is important to have a proxy for tracking this capability which could one day result in automation or malicious code generation. The APPS benchmark can provide an important measure for tracking upstream program synthesis advancements.

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
