# OpenReview forum: "Measuring Coding Challenge Competence With APPS"
_NeurIPS.cc/2021/Track/Datasets_and_Benchmarks/Round2 — NeurIPS 2021 Datasets and Benchmarks Track (Round 2)_

### Official Review · Reviewer_DPaD · 2021-09-18
**Valuable benchmark to evaluate LMs of code, but does not cover various software engineering tasks.**

**Rating:** 7
**Confidence:** 4
**Correctness:** The claims made in the submission are…
**Clarity:** The paper is well written and easy to…

**Strengths:**

The paper introduces an important benchmark to evaluate code generation. The authors correctly point out several limitations of traditional NLP based evaluation metrics like BLEU score when applied to code, and provide solution.

Authors correctly point out, that a passed unit tests does not guarantee program correctness as there might be false positives. The propose multiple unit tests, and evaluate the likelihood of false positives.

It is important that APPS benchmark is not geared towards a specific model architecture (e.g., GPT) or a modeling task (language modeling). Or work only with left-to-right code generation models. While large language models are SOTA right now, their performance is still very low (up to 20-30% accuracy). It is likely that a new technology will emerge (perhaps, it will utilize programming language grammars, and may be based on an encoder-decoder architecture, and not left-to-right code generation) which will outperform those. It is important that the APPS benchmark remains applicable to those.

**Weaknesses:**

Since the APPS dataset is a collection of problems extracted from public websites like CodeForces, there is a risk of data leakage, especially since modern language models of code (e.g. Codex) are trained on a large fraction of GitHub and may have seen it in the training set.

The APPS benchmark targets a narrow task of code generation from natural language description. There are numerous other software engineering tasks (involving code gen) that may not be covered by this benchmark. For instance, program repair task which is generally formulated as follows: given a buggy code produce a fixed code. Or code summarization – which is an inverse, code to NL task. Can these types of problems be formulated the same way? They can be solved with language models.

There exist similar benchmarks: HumanEval. It would be good to understand the difference.

Real-world code generation should take into account file-level (import statements, class and method signatures) and even inter-procedural contexts. As in the real interview, candidate might be allowed to use numpy to implement a given algorithm. But generally it might be interesting to consider code generation given specific source code project/library.

**Additional Feedback:**

Figure4 label is confusing: It shows the size of the respective GPT-Neo models, but states “few-shot” for GPT-3 instead of the size.

L308: “Open AI Codex is a 12B parameter Transformer language model” typically, Vaswani’s encoder-decoder transformer model is referred to as the Transformer, which incorrectly suggests that Codex is an encoder-decoder model, while Codex is the GPT-style model (decoder-only transformer). While language modeling is possible with encoder-decoder models, throughout the task you distinguish two models (encoder decoder Transformer and GPT) so might as well do it here.

**Documentation:**

The documentation seems clear, but extra inputs from authors/tutorials might be needed to use it.
Is there an online platform to facilitate APPS evaluation/running unit tests?

**Ethics:**

No ethics concerns.

**Relation To Prior Work:**

There exist similar benchmarks: HumanEval. It would be good to understand the difference. Both benchmarks appeared at similar time, so I am not sure it should be counted as prior work.

**Summary And Contributions:**

I would like to thank authors for their responses.
I am updating based on the responses.

The paper introduces a dataset for evaluating code generation from natural language specifications.
The evaluation setting aims to mirror how human programmers are evaluated during an interview: given a problem description, their task is to implement an algorithm that solves the problem. The correctness of generated programs is assessed via unit tests. The dataset consists of 10000 samples in Python programming language, which are subdivided into 3 distinct difficulty categories.

The APPS benchmark can potentially serve as an important and valuable asset in evaluating language models of code, which targets a real-world informative setting, and overcomes several limitations of the benchmarks which are based on string matching and n-gram based criteria like accuracy, BLEU or ROUGE.

The APPS benchmark targets a narrow task of code generation from natural language description. There are numerous other software engineering tasks (involving code gen) that may not be covered by this benchmark. For instance, program repair task which is generally formulated as follows: given a buggy code produce a fixed code. Or code summarization – which is an inverse, code to NL task. Can these types of problems be formulated the same way? They can be solved with language models.

---

> ### Author Response · Authors · 2021-09-29
> **Response to Reviewer DPaD (1/2)**
>
> Thank you for your careful analysis of our work. We hope the following response addresses your concerns, and we hope you will champion our paper.
>
> **The memorization hypothesis cannot predict our results.**
>
> We agree that data leakage is a potential problem for models that do not train on decontaminated data (as we do for our GPT-2 models). However, we do not believe this is an issue in practice. As we mention in the paper, memorization explaining all performance is an implausible explanation as performance tracks problem difficulty. If memorization were a practical issue, we would expect to see uniform performance across difficulties or higher performance in interview problems than introductory (since solutions to interview problems are more readily available online). However, we observe the opposite: strict accuracy tracks with difficulty for all models we evaluate and for OpenAI Codex models.
>
> **Clarifying the scope of APPS.**
>
> We agree that the problem of generating code from natural language specifications is only one aspect of code generation. With APPS, we hope to foster new work into this important subproblem that was previously lacking a rigorous benchmark and is well-suited for further investigation with the advent of large language models.
>
> Your suggestion to incorporate evaluations under different settings (e.g. where certain import statements are allowed) is interesting and novel to best of our knowledge. This would be a good direction for future work. In APPS, we allow models to access standard Python libraries via import statements. We will clarify this in the updated paper.
>
> **Comparisons to additional benchmarks.**
>
> Since the release of APPS, two additional code generation datasets have been released that are similar in nature: HumanEval [1] and MBPP [2]. Compared to the 10,000 problems in APPS, these have 164 and 974 problems respectively (with training set sizes of 0 and 374 compared to 5,000 in APPS). However, we view these datasets as complementary to APPS. HumanEval is unique because it only consists of handwritten problems, and MBPP focuses on introductory level problems, where the current models are making the most progress. We will expand on these comparisons in the updated paper.
>
> [1]: Chen et al. “Evaluating Large Language Models Trained on Code”. arXiv preprint arXiv:2107.03374, 2021
>
> [2]: Austin et al. “Program Synthesis with Large Language Models”. arXiv preprint arXiv:2108.07732, 2021

---

> ### Author Response · Authors · 2021-09-30
> **Response to Reviewer DPaD (2/2)**
>
> **Additional responses.**
>
> We have extensive documentation inside the GitHub repository, including step-by-step instructions for running code. In the future, we may create a CodaLab leaderboard for a future competition were the paper to be accepted.
>
> We will adjust the terminology in the updated paper so that GPT-2 and GPT-Neo are clearly distinguished from encoder-decoder models, thanks to your comment.

---

### Official Review · Reviewer_7Lcy · 2021-09-20
**Although small size, but good effort on measuring Coding Challenge Competence that debunks some myths/hype.**

**Rating:** 7
**Confidence:** 5
**Clarity:** Paper is well written and clear!

**Strengths:**

Goal is good – with so much hype around the “super models”, it is important to have a high-quality or any reasonable dataset to evaluate them.
Curation effort is good – the appropriate cleansing such as problem cleanup, de-duplication, and test case standardization, and decontamination is in place
APPS is also used by the Codex paper.
The biggest contribution of the paper is to show that GPT-3, with all its hype, really cannot generate code with consistency. You cannot reap what you did not sow – since GPT-3 is not trained on code, it is not good at that.


**Weaknesses:**

The dataset is way too small to assess whether these large language models can do code generation well. It is not clear whether the extreme low strict accuracy is the result of a small fine tuning training set (5000 data samples) or the weakness of the neural network model. It would be very interesting to use sequentially bigger fine tuning training set (10000, 20000, 50000, 100K samples, etc) to see if the strict accuracy keeps increasing or saturates.
Limitation of the dataset is not discussed. The scope (pedagogical problems) and size (5000 training) of the dataset is too limited to thoroughly evaluate the problem of code generation. Granted that the goal is ambitious, and it is a good start, a discussion of the limitations is warranted.
The list of observations in the paper are rather obvious, and I do not agree with observation 3. The reduction of syntax errors is due to pre-training of the GPT-2 and GPT-Neo with code instead of just texts, so they learned the programming language model better.


**Additional Feedback:**

Show some examples of successful code generation to see how simple they are (e.g. are they one-liner or more complex). Are the generated code examples in the supplementary correct code?
Show the performance metrics for the pure pre-trained (not fine-tuned) GPT-2 and GPT-Neo models to get a baseline and an indication of how much fine-tuning helps.
Use sequentially bigger fine tuning training set (1000, 2000, 5000 samples, etc) to see how the strict accuracy reacts to fine tuning training dataset size


**Correctness:**

Please refer to the weakness section above! Evaluation methods and experiment design has been appropriately performed!

**Documentation:**

The construction and curation of the dataset is appropriately discussed

**Ethics:**

CodeForces codes and permissions for use in a dataset could be a concern and authors should seek a legal subject matter expert review on it, This is always done in industry as due diligence and will be important for this dataset.

**Relation To Prior Work:**

Section on “Related Work” is too heavily focused on discussing code generation technique in prior art. This paper is more about the dataset and less about code generation technique, so there should have been more emphasis on comparing dataset quality with prior datasets with the same goal.

**Summary And Contributions:**

The authors curated a dataset, APPS, as a benchmark for evaluating the quality of machine learning models for code generation/synthesis from a natural language (English) description of the coding problem.
The dataset is collected from online judging systems (CodeForces, AtCoder, etc.) and other coding competitions (ACM, USACO, IOI, etc.) and contains 10000 coding problems, each with its problem description, test cases with specific input-output pairs, and some ground-truth (correct) solutions implemented in Python. Problems are classified into introductory level (3639), interview level (5000), and competition level (1361).
10000 problems, 131,777 test cases, 232,412 correct solutions (Python), average length of problem description is 293.2 word.
Curation:
wrote custom HTML parsers for each source of problems, which allows us to properly format LaTeX expressions, lists, and sections in the question text.
convert equation images to LaTeX using theMathPix API
perform deduplication using tf-idf features with SVD dimensionality reduction and cosine similarity
Several graduate and undergraduate students worked for 6 months

Test cases evaluation:
merges the judging functionality of several judging systems also standardize the format of test cases.

Experimental setup:
Pre-train GPT-2 on GitHub (decontaminated) – 30 GB of Python code
Use GPT-3 Instruct model
Use pre-trained GPT-Neo (pretrained PILE which included GitHub) and fine tune
Table 2 summarizes all the results – Strict accuracy is the honest metric.
Observation and analysis of the authors:
1.	Models can sometimes generate correct or superficially plausible code
2.	Many generated programs are free of syntax errors
3.	Syntax errors decrease exponentially with fine-tuning and increased model sizes.
4.	Using top-5 improves performance metrics
5.	GPT-3 does not do well
6.	BLEU scores are not a good measure of model quality for code generation
7.	The Codex model does better, but far from have solved code generation

Main Conclusions:
GPT-3 performs quite badly for code generation
Some neural network models pre-trained on code can start to do code generation for some extremely simple programming problems, so there is a long, long way to go.

---

> ### Author Response · Authors · 2021-09-29
> **Response to Reviewer 7Lcy**
>
> Thank you for your careful analysis of our work. We hope the following response addresses your concerns.
>
> **APPS is the largest benchmark for complex code generation from NL specifications.**
>
> We compare APPS with numerous existing code generation benchmarks in Table 1 of the main paper and the Supplementary Material. To the best of our knowledge, APPS is by far the largest benchmark with test cases for generating complex code from natural language specifications. Since the release of APPS, two similar benchmarks have been proposed: HumanEval [1] and MBPP [2]. Compared to the 10,000 problems in APPS, these have 164 and 974 problems respectively (with training set sizes of 0 and 374 compared to 5,000 in APPS).
>
> Furthermore, while APPS has 5,000 programming problems in the training set, the number of actual examples that models train on is 232,412, i.e. the number of ground-truth solutions. This is far larger than most fine-tuning datasets, and the upshot can be seen in Table 2 of the paper: GPT-2 0.1B fine-tuned on APPS outperforms GPT-3 175B without fine-tuning. Thus, fine-tuning on APPS makes up for a difference in parameter count of 3 orders of magnitude.
>
> **Clarifying the scope of APPS.**
>
> APPS is intended to measure a very specific aspect of code generation, namely whether models can convert arbitrary natural language specifications into open-ended Python code satisfying these specifications. While this is more ambitious than previous tasks such as pseudocode-to-code translation, we agree that there are many aspects of code generation that APPS does not evaluate. We will clarify this point in the updated paper.
>
> **APPS is fully legally compliant.**
>
> To prevent confusion, we have augmented the analysis of our legal compliance in the Supplementary Material. Please see the relevant section for details on how APPS is fully legally compliant.
>
> **Comparing syntax errors across models.**
>
> We agree that differences in the initial pretraining of the models we evaluate makes certain inferences harder. Even though we cannot precisely disentangle the effects of training dataset and model size, we do show that fine-tuning on APPS makes a substantial difference in syntax errors compared to GPT-3 175B, and using a larger model pretrained on more code increases performance yet again. We will clarify this in the updated paper.
>
>
> [1]: Chen et al. “Evaluating Large Language Models Trained on Code”. arXiv preprint arXiv:2107.03374, 2021
>
> [2]: Austin et al. “Program Synthesis with Large Language Models”. arXiv preprint arXiv:2108.07732, 2021

---

### Official Review · Reviewer_qKG4 · 2021-09-20
**Novel dataset and benchmark with some weaknesses**

**Rating:** 6
**Confidence:** 3
**Correctness:** See second point in the Weaknesses se…

**Strengths:**

The principal strengths of the paper are:
1. Compared to the restricted settings of the previous work on code generation, the APPS dataset can be used to learn the task of generating full fledged Python code given the textual problem specification of competitive coding challenges.
2. Instead of using BLEU to evaluate the generated solutions, the dataset contains test cases that can be employed as a gold-standard metric for correctness.

**Weaknesses:**

The major weaknesses of the paper are:

1. The dataset does not decouple the task of algorithm design and code generation. One cannot be sure if a language model has not simply memorized the algorithmic implementations. Line#265 argues that memorization would follow uniform performance across all categories. Instead, we might expect a memorizing model to do well on those category of problems it has seen more of. For instance, GPT-Neo was pre-trained on contaminated data from Github. It is likely that Github has large amounts of example code for introductory problems, followed by some solutions to the interview level problems and even fewer solutions to competition level problems. This is an alternate plausible explanation for GPT-Neo's performance across the difficulty levels.

2. The paper makes several doubtful claims. The abstract mentions that GPT-Neo can pass approximately 20% of the test cases of introductory problems. However, only in the final Experiments section is it mentioned that GPT-Neo under went pre-training on contaminated Github data. In general, all the models being compared and analyzed side-by-side undergo very different training regimes. GPT-2 is pre-trained on decontaminated Github data and then fine-tuned on APPS. GPT-3 is only used in the few-shot setting. The results from these experiments cannot be used to make sweeping claims such as in line#60: "We find that accuracy decreases with difficulty level and improves through fine-tuning and model size increases." In Table 2, we see that only the accuracy of GPT-Neo decreases with difficulty level. For all other models, the accuracy increases when going from introductory level to interview level and then decreases from interview level to competitive level.

**Additional Feedback:**

-

**Clarity:**

The paper is well-written and easy to follow.
Except in lines 183 and 184 where the false positive rate is calculated using some quantities like 890 and 765 which are not clearly defined.

**Documentation:**

The paper and supplementary material contain adequate details on dataset creation process. The license and datasheet are also provided.

**Ethics:**

No, there does not seem to be any ethical concerns.

**Relation To Prior Work:**

The literature survey is exhaustive and the proposed work is differentiated well from all the related work.

**Summary And Contributions:**

The authors develop the APPS dataset for code generation from natural language specifications. It consists of 10k examples categorized into three levels of difficulty. Each example consists of a coding problem specification, one or more coded solutions written by humans and one or more test cases to check the generated solution. The authors evaluate state-of-the-art language models on the dataset to find a large room for improvement.

The contributions of the paper are as follows:
1. Novel benchmark for code generation from natural language specifications
2. Collection of 10k examples from open-access competitive coding websites
3. Evaluation of generated solutions using test cases instead of using BLEU

---

> ### Author Response · Authors · 2021-09-29
> **Response to Reviewer qKG4**
>
> Thank you for your careful analysis of our work. We hope the following response addresses your concerns.
>
> **The memorization hypothesis cannot predict our results.**
>
> Due to incentives for passing coding interviews, websites specializing in interview-level problems such as LeetCode are far more popular than sources we use for introductory problems, such as Codewars. For instance, a randomly selected LeetCode problem (1956 in the training set) has 287,531 accepted solutions, while a randomly selected Codewars problem (4689 in the training set) has 193 accepted solutions and is substantially easier. Thus, if memorization were a practical issue, we would actually expect accuracy on interview-level problems to be highest, but this is not what we observe. Furthermore, many of our test problems are from Kattis, which has no publicly available solutions.
>
> **Improved clarity in discussion of performance trends.**
>
> In line 60, we meant to convey that strict accuracy is decreasing with difficulty level, which occurs for all models we evaluate (including GPT-3 few-shot). This also holds for OpenAI Codex [1], a much larger model trained on significantly more data. We will add this clarification to the updated paper.
>
> We agree that GPT-Neo 2.7B’s pretraining dataset may be responsible for some of its performance improvements over GPT-2 1.5B. This is mitigated by the fact that we pretrain GPT-2 on GitHub code as well, but it is difficult to fully disentangle the effects of model size and training dataset across these two works. Additionally, GPT-Neo’s architecture differs from GPT-2. In the updated paper, we will clarify these potential confounders that may explain the observed correlation between model size and accuracy. If we addressed the thrust of your concerns, we kindly ask that you consider raising your score.
>
> [1]: Chen et al. “Evaluating Large Language Models Trained on Code”. arXiv preprint arXiv:2107.03374, 2021

---

### Official Review · Reviewer_kWe4 · 2021-09-21
**Exciting results and an important but SLOPPY dataset**

**Rating:** 6
**Confidence:** 3
**Correctness:** I have not re-run their experiments.
**Clarity:** Yes

**Strengths:**

* The dataset consists of 10k problems in what seems to be well-formatted English problems (some with latex), many accompanied by Python solutions and test cases.
* The models can actually solve some of the test cases and the code they generate looks very interesting.
* The paper is well-written.
* The dataset has already been used in notable follow-up work on arXiv.

**Weaknesses:**

The paper presents cherry-picked statistics. For example, they will state that the average number of tests per problem is large but do not mention that more than half the test problems have very few test cases, and most of those test cases are directly written as examples in the Input-Output problem. Contrary to what the authors think, presenting one's own weaknesses does not ruin a paper---in fact it makes it much stronger.

**Additional Feedback:**

Given the above mistakes found in the dataset, how should we trust that other things like de-duplication/etc. were done carefully. For example, a simple test would be to run the solutions from one problem on all others -- one would not expect the solution to a non-trivial problem to solve another non-trivial problem. Have you tried this? Did you find that any solutions to some problems solve others? This paper does not seem to discuss the checking and double-checking that should go into creating such a valuable resource.

**Documentation:**

The detail in the description of how the dataset could be improved, hopefully it will be included in greater detail in an appendix.

**Relation To Prior Work:**

Yes

**Summary And Contributions:**

This paper introduces an exciting dataset of 10,000 programming problems in English with test cases to evaluate correctness. The problems are scraped from various websites. An evaluation of language models shows that GPT models, appropriately fine-tuned, can write code that solves some of these problems. This is quite exciting since it has been very hard for computers to solve non-trivial problems described in natural language. This dataset is of significant size and quality that could be useful for training and evaluation of program synthesis systems.  This is a good paper -- if were been written more objectively and acknowledged it's limitations, it would be a great paper.

---

> ### Author Response · Authors · 2021-09-29
> **Response to Reviewer kWe4 (2/2)**
>
> **Creating APPS was a substantial undertaking.**
>
> Creating APPS required several months of manual cleaning, which was a substantial undertaking that by its nature would be difficult to replicate. We describe the main steps of our 6-month curation process in Section 3 of the paper, including the most relevant details for users of APPS, and people can easily build on APPS using our problem formatting. However, it is unheard of to have a single script that can reproduce a dataset’s collection. We provide source URLs (which most datasets do not, such as ImageNet) and provide details in the paper for reproducibility. The only part that is not reproducible with a script is the several months of manual cleaning required to create APPS.
>
> **Quality control and correctness.**
>
> To create APPS, a team of authors performed extensive quality control, including a standard deduplication procedure and removing problems that require images, as we mention in Section 3 of the paper. We identified duplicate problems using pairwise cosine similarities between tf-idf features computed on problem statements using numpy and sklearn, which resulted in a small set of duplicates that we were able to manually verify and remove. We are confident that this was done correctly.
>
> Problem 500 in the test set is actually from https://codeforces.com/problemset/problem/761/E and contains a purely “fun” image (the beginning text for the two problems is similar, which may have led to the mixup). In fact, problem 761_F was not included in APPS. During the process of converting images to LaTeX, we found that the vast majority of images in Codeforces problems are purely illustrative and are not necessary for solving the problems. In cases where images may have been necessary, we remove the problems from APPS.
>
> Approximately 7% of the problems in the APPS test set do admit multiple solutions, nearly all of which are from Codeforces. Thank you for pointing out this issue. By manually checking 100 test cases across 10 randomly sampled problems from this set, we estimate that 3.7% of the test cases and 4.2% of the problems in APPS are affected in practice (test cases are deemed unaffected in this preliminary investigation if three ground-truth solutions agree on the single provided output that we originally scraped). Note that this issue can only cause performance to be underestimated; current models are performing better than we thought they were. We will fix this issue as soon as possible with an updated version of APPS. This will enable future evaluations on APPS to remove the downward bias on performance brought about by these test cases. Thank you again for pointing out this issue, which in practice only affects a small number of test cases but is nevertheless important.
>
> **Collection of test cases.**
>
> We collect test cases from publicly available access points on the websites we scrape from. That is, all the data we scrape can be readily viewed online and we never scrape from behind paywalls. For instance, ground-truth solutions and test cases for Codeforces are aggregated from competition pages on the Codeforces website, such as https://codeforces.com/contest/299/status/A using a custom HTML scraper. We will clarify this in the updated paper.
>
> **Code generation is not the same as multiple-choice QA.**
>
> In multiple-choice QA tasks, models typically output a categorical distribution that is evaluated against a single ground-truth answer. The task in APPS of generating arbitrary Python code from natural language specifications is wholly different.
>
> The output space of models trained on APPS is not a single answer, but rather Python code that is evaluated on multiple test cases that may themselves follow a wide variety of formats. Understanding and adhering to the correct input and output formats is itself a nontrivial task for arbitrary Python coding challenges and cannot be gamed by random guessing. For instance, while the set of valid outputs for a test case may be binary, after 34 characters the set of possible ASCII strings is larger than the number of atoms in the observable universe (the average length of APPS solutions is 510 characters).
>
> Further, the majority of questions in APPS have many test cases, such that once models learn to generate code that follows formatting instructions, they still cannot game our accuracy metric by generating code that randomly outputs elements of the solution space. Even if all test cases were binary, the average APPS problem with 21 test cases would have a 1 / 2,097,152 chance of being a false positive. If models use the 14-way strategy, it would have a 1/14^21 = 8.5 * 10^-25 chance, The suggested 14-way guessing strategy would not beat existing models at either test case average or strict accuracy.
>
> We hope our response clarifies the substantial effort and quality control that went into the construction of APPS. If we addressed the thrust of your concerns, we kindly ask that you consider raising your score.

---

> > ### Comment · Reviewer_kWe4 · 2021-09-30
> > **Further issues with the input-output cases**
> >
> > # Test cases from English descriptions
> >
> > I just examined problem 500 more closely. First, like the other problem I checked, *all* the test cases from the English description are included in input_output.json file!
> >
> > What I mean is that a problem says
> > "Some english description
> >
> > example_in_1, example_out_1
> > example_in_2, example_out_2
> > example_in_3, example_out_3"
> >
> > Then those three examples make up the first three examples in input_output.json.
> >
> > I suspect this is the case for many/all of the problems, but I don't want to spend my time checking. How many of the test cases are directly from the English description and how many are novel? And how many problems have test cases that are only from the English descriptions? Could you at least indicate that in the json input_output files? Even if *your* baselines GPT-Neo don't "cheat" and extract examples from the English text, the paper is about the *dataset*. If another system can beat yours by extracting the sample input/outputs from the English descriptions, then it is not a great program synthesis dataset.
> >
> > # Lack of good test cases from Kattis
> >
> > You write:
> >     "Thus, we had to make do with the publicly available test cases, which are
> >     also the test cases provided as examples in problem statements."
> >
> > This is *not true*. If you cannot extract high-quality test cases from Kattis problems, you could chose not to include them in the dataset. Or, you could include them but mark them clearly with this issue.
> >
> > # Reproducibility
> >
> > I disagree with your comment that reproducibility does not matter just because one cannot make a script to reproduce your dataset exactly. First, even if the exact dataset cannot be reproduced, documenting the steps taken is important for reproducing similar datasets. (See the importance of the dataset ImageNetV2 which the authors took great care to create in the same way as ImageNet.) Second, if your steps were done by a sequence of scripts, they may be largely reproducible, or at least publishing those scripts may be helpful. If you manually went through the problems then it would be useful to know roughly what criteria you used in this manual process.
> >
> > # Discussing limitations
> >
> > Generally speaking, I feel like your paper does not bring forward the issues with the dataset. I suspect that you were aware of the issues (mostly missing URLs, test cases from English descriptions in input_output). A carefully written paper would bring those issues (and other issues that I may not have discovered in my quick evaluation) forth in the writeup.
> >
> > # Adding URLs
> >
> > I applaud you for adding URLs. (It seems like I was correct you have test problem 500 as 761/E not 761/F, so I am not sure what the confusion in your rebuttal is about.)

---

> > > ### Public Comment · ~Leonard_Tang1 · 2021-09-30
> > > **Quick Check of Input-Output**
> > >
> > > I've been using APPS, and I just checked problem 500. It has 3 examples in the problem statement, but it also has 48 additional unseen test cases.
> > >
> > > I think it's common for programming practice sites (e.g. LeetCode, HackerRank), to test solutions with problem statement examples and additional (unseen) examples.

---

> > > ### Author Response · Authors · 2021-09-30
> > > **Response to Reviewer kWe4**
> > >
> > > As noted by the commenter, coding sites use problem statement examples in their test cases. We followed the precedents adhered to on the sites from which we collected examples. In fact, these additional test cases decrease the probability that models could succeed on the Strict Accuracy metric. This is why including these test cases on Kattis problems helps improve the quality of evaluation. Followup works such as OpenAI Codex do not evaluate on Test Case Average and only use Strict Accuracy, following our suggestion in the paper (line 233). We will further augment the discussion of test cases and strict accuracy thanks to your comment.
> > >
> > > We agree that reproducibility is important, and this is why we provide details for creating APPS in Section 3. Since we provide URLs, the dataset should be easy to reproduce, save for the several months of manual cleaning we underwent. We did not originally include URLs as there is not a precedent for including URLs for individual examples in datasets. For instance, ImageNet does not include original Flickr URLs. However, we agreed that the URLs could provide value and thus included URLs for all problems thanks to your suggestion. We hope our response clarifies our design choices and the precedents we followed. If we addressed some of your concerns, we kindly ask that you consider raising your score.

---

> ### Author Response · Authors · 2021-09-29
> **Response to Reviewer kWe4 (1/2)**
>
> Thank you for your careful analysis of our work. We hope the following response addresses your concerns.
>
>
> **All problems now have source URLs.**
>
> The lack of source URLs for some problems was a point of concern for us as well. We have been working on addressing this and have now updated APPS with source URLs for all problems in the training and test sets.
>
> **Similarity of training and test distributions.**
>
> As you point out, there are some differences in the training and test sets. This is due to careful design choices that improve the quality of APPS:
> - Since not all problems that we collect have public test cases, we place problems with higher numbers of test cases in the test set.
> - In order to have more competition-level problems in the dataset, we place all Kattis problems in the test set. This was necessary, because 50.6% of the competition problems are from Kattis, and these could not be used for the training set (note that this explains the discrepancy in availability of URLs that you mention).
>
> On important metrics, our training and test sets are similar: both contain a significant number of problems from each difficulty, and both contain problems from a diverse set of sources. One difference you point out is the distribution shift in the number of STDIN and Call-Based problems: the training set has a 36/64 split while the test set is mostly STDIN problems. This single source of imbalance enabled meeting our other more valuable criteria, and does not present an issue in practice, as modern models are routinely evaluated on different data than they were trained on. Moreover, all models trained on APPS are affected by this imbalance equally, so it does not affect comparisons between methods.
>
> **APPS is fully legally compliant.**
>
> To prevent confusion, we have augmented the analysis of our legal compliance in the Supplementary Material. Please see the relevant section for details on how APPS is fully legally compliant.
>
> **Necessary design choices for Kattis (test cases in problem 3500).**
>
> The issue you point out with problem 3500 is due to a necessary choice we made for Kattis problems. While Kattis has thousands of high-quality coding challenges, it has very few public test cases. We were in contact with Kattis for over a month, but due to logistics they were not able to share more test cases. Thus, we had to make do with the publicly available test cases, which are also the test cases provided as examples in problem statements. To give models ample context while also preserving the integrity of evaluation, we opted to include all available test cases in the input_output.json files while displaying all but one test case in the question text. This is a good middle ground given the nature of the models we are evaluating. Namely, models do not hard-code input-output pairs from the problem text, as they are trained to generate legitimate solutions. Even if they were to do so, gaming the strict accuracy metric is prevented via the held-out test cases.
>
> Aside from visual inspection, we can know that models do not hard-code solutions in practice by comparing accuracy on visible and held-out Kattis test cases. For a GPT-Neo 2.7B model, these are 6.95±3% and 6.23±3.8% respectively (alpha=0.05 confidence intervals computed using Hoeffding’s inequality). Thus, there is no significant difference in accuracy, implying that models do not hard-code solutions in practice. This validates our design choice for curating Kattis test cases.

---

### Comment · Reviewer_kWe4 · 2021-09-23
**Multiple-choice coding challenge?**

I played a little more with the dataset.

Unless I'm mistaken, it seems that over 40% of the the test case solutions y are in {0...10, -1, yes/no}

So if you can do well on a 12-way multiple choice test, you may be able to do well on the dataset

Here is the command I ran:

```python
>> mean([str(y).strip().lower() in "-1 0 1 2 3 4 5 6 7 8 9 10 yes no" for xy in input_output for y in xy["outputs"]])
0.43534307850557247
```

The > 40% rate holds across competition/interview/introductory levels.

This seems relevant to the 20% test case accuracy touted in the abstract, but I did not see such statistics discussed in the paper.

---

> ### Author Response · Authors · 2021-09-30
> **Writing Code Is Not a Multiple-Choice Problem**
>
> First, we would like to clarify that this is not a multiple-choice task. This is a code generation task. The output spaces are entirely distinct; one has a combinatorially large output space, the other does not.
>
> In multiple-choice QA tasks, models typically output a categorical distribution that is evaluated against a single ground-truth answer. The task in APPS of generating arbitrary Python code from natural language specifications is wholly different. The output space of models trained on APPS is not a single answer, but rather Python code that is evaluated on multiple test cases that may themselves follow a wide variety of formats. Understanding and adhering to the correct input and output formats is itself a nontrivial task for arbitrary Python coding challenges and cannot be gamed by random guessing. For instance, while the set of valid outputs for a test case may be binary, after 34 characters the set of possible ASCII strings is larger than the number of atoms in the observable universe (the average length of APPS solutions is 510 characters).
>
> Further, the majority of questions in APPS have many test cases, such that once models learn to generate code that follows formatting instructions, they still cannot game our accuracy metric by generating code that randomly outputs elements of the solution space. Even if all test cases were binary, which they are not, the average APPS problem with 21 test cases would have a 1 / 2,097,152 chance of being a false positive. If a model were to use the _14_-way strategy, it would have a 1/14^21 = 8.5 * 10^-25 guessing probability. The suggested _14_-way guessing strategy would not beat existing models at either test case average or strict accuracy.

---

### Decision · Program_Chairs · 2021-10-09

**Decision:**

Accept

**Comment:**

The paper proposes a dataset of 10k programming problems to promote research in code generation. The authors then evaluate SOTA language models on the proposed benchmark. All the reviewers agree that the proposed dataset is a good contribution. Therefore, I recommend acceptance of this paper.